# *Brucella* and Its Hidden Flagellar System

**DOI:** 10.3390/microorganisms10010083

**Published:** 2021-12-31

**Authors:** Roberto F. Coloma-Rivero, Manuel Flores-Concha, Raúl E. Molina, Rodrigo Soto-Shara, Ángelo Cartes, Ángel A. Oñate

**Affiliations:** Laboratory of Molecular Immunology, Department of Microbiology, Faculty of Biological Sciences, Universidad de Concepción, Concepción 4030000, Chile; rcoloma@udec.cl (R.F.C.-R.); manuelflores@udec.cl (M.F.-C.); ramolina@udec.cl (R.E.M.); rodrsoto@udec.cl (R.S.-S.); angelocartes@udec.cl (Á.C.)

**Keywords:** *Brucella abortus*, flagellum, flagellin, ORF, virulence factors

## Abstract

*Brucella*, a Gram-negative bacterium with a high infective capacity and a wide spectrum of hosts in the animal world, is found in terrestrial and marine mammals, as well as amphibians. This broad spectrum of hosts is closely related to the non-classical virulence factors that allow this pathogen to establish its replicative niche, colonizing epithelial and immune system cells, evading the host’s defenses and defensive response. While motility is the primary role of the flagellum in most bacteria, in *Brucella,* the flagellum is involved in virulence, infectivity, cell growth, and biofilm formation, all of which are very important facts in a bacterium that to date has been described as a non-motile organism. Evidence of the expression of these flagellar proteins that are present in *Brucella* makes it possible to hypothesize certain evolutionary aspects as to where a free-living bacterium eventually acquired genetic material from environmental microorganisms, including flagellar genes, conferring on it the ability to reach other hosts (mammals), and, under selective pressure from the environment, can express these genes, helping it to evade the immune response. This review summarizes relevant aspects of the presence of flagellar proteins and puts into context their relevance in certain functions associated with the infective process. The study of these flagellar genes gives the genus *Brucella* a very high infectious versatility, placing it among the main organisms in urgent need of study, as it is linked to human health by direct contact with farm animals and by eventual transmission to the general population, where flagellar genes and proteins are of great relevance.

## 1. Introduction

*Brucella abortus* is a Gram-negative, non-motile, coccobacillary bacterium and a facultative intracellular pathogen, causing disease in both animals and humans [1]. Brucellosis, the name given to the disease caused by this pathogen, is a zoonosis described in cattle and other mammals. Being an opportunistic agent, it can infect humans, and diagnosis is difficult [2]. This disease can be characterized by its ability to become a chronic infection, which in bovines can cause abortions, stillbirths, the birth of weak calves, and sterility in males [3]. It is endemic in many areas of the world, and, in humans, the main sources of infection are through the consumption of contaminated meat and dairy products and through contact with secretions from infected animals [2,4,5]. *Brucella* enters the host organism through traumatized skin (wounds) or through the air to the mucosa in the form of an aerosol [6]. Once inside the host organism, *Brucella* spreads and multiplies in the lymph nodes, spleen, liver, bone marrow, mammary glands, and sexual organs via macrophages [7]. *Brucella* can compromise the gastrointestinal, hepatobiliary, genitourinary, skeletal muscle, cardiovascular system, and integumentary systems [8], which can become a chronic condition, crossing the blood-brain barrier through infected monocytes like a Trojan horse and achieving re-infection in the midbrain, causing neurobrucellosis [9]. If the bacteria are not killed, they can survive for long periods inside these phagocytic cells, in phagosomes, where they can multiply by inhibiting fusion with the lysosome through rapid acidification into the *Brucella*-containing vacuoles (BCVs) [4,7,10,11]. In non-phagocytic cells, *Brucella* tends to localize within the rough endoplasmic reticulum [2,12] where, like other intracellular bacteria, it survives in the host cell. The genus *Brucella*, identified in 1920 by Meyer and Shaw, currently comprises twelve species differentiated by their tropism, pathogenicity, and host phenotypic traits [13]. A decade ago, the genus contained six “classic” *Brucella* species (*B. melitensis*, *B. abortus*, *B. suis*, *B. canis*, *B. ovis,* and *B. neotomae*), which are also known as the “core” *Brucella*. Three of these species (*B. melitensis*, *B. abortus* and *B. suis*) are the pathogens that cause one of the most significant zoonoses worldwide [14]. The group of classic *Brucella* species was expanded in 2007 to include *B. ceti* and *B. pinnipedialis*, bacteria isolated from marine mammals [15]. Currently, new technological strategies like massive genetic sequencing are being used to group certain organisms with characteristics of the genus *Brucella* (wherein about 30 species and other serovars not previously described have been included) [16].

Due to its low infectious dose and easy transmission in the form of an aerosol, *Brucella* was one of the first microorganisms to enter consideration for use as a biological weapon, classified by the United States Army in the 1950s [17], and cataloged by the Centers for Disease Control and Prevention (CDC) and the National Institute of Allergy and Infectious Diseases (NIAID) as a Category-B bioterrorism agent [14]. Emphasizing the genes that command its infectious cycle, the study of *Brucella* offers a clearer view of its infective process and pathogenesis, and, in turn, a better understanding of how this genus establishes its replicative niche in host cells and can perpetuate its chronicity. Some years ago, an interesting finding was made regarding this bacterium, generally classified as non-motile, in terms of the expression of flagellar genes and a polar flagellum under certain conditions.

## 2. Bacterial Flagellum

The bacterial flagellum is a huge molecular complex made up of 20,000 to 30,000 protein subunits of approximately 30 different proteins [18]. The structural components of the flagellum can be divided into two parts: the basal body rings and the tubular axial structure [19]. The rings of the basal body form the rotary motor, together with the stator complex, composed of the cytoplasmic membrane proteins MotA and MotB [20,21]. The basal body rings consist of four main ring structures—the L ring, the P ring, the MS ring, and the C ring—where the latter two form the rotor–stator complex [22,23]. The latter complex is coupled to the proton flux, and all of this, in turn, surrounds the transporter gate formed by the FliE, FlgB, FlgC, and FlgF proteins [24]. The LP ring acts as bushing, supporting the distal rod in its rapid and stable rotation without much friction [25], which leads to flagellar movement [20,26]. Likewise, the axial structure consists of three main parts: the rod, the hook, and the filament [27,28]. The rod is a transmission shaft with an approximate length of 30 nm, which connects the rotor rings and the hook; this rod is a fairly complex helical cylinder composed of four flagellar proteins: FlgB, FlgC, FlgF, and FlgG [29,30]. The hook is a short, curved segment, approximately 55 nm long, which is made up of a helical set of subunits of the hook’s own flagellar protein (FlgE) and acts as a universal joint structure that transmits movement to the filament, regardless of its orientation. Between the filament and the hook, there is a short axial structural segment with a thinner and smoother appearance, called the hook and filament junction, formed by the adapter proteins FlgK and FlgL [24]. The filament is a thin helical structure of approximately 20 nm in diameter and typically grows to about 15 μm long, composed of approximately 20,000 flagellin protein (FliC) subunits [31,32,33]; it acts as a rigid helix to produce thrust for the cell to swim in aqueous environments [34,35,36]. This filament ends at its apex with the Cap protein (FliD), which forms a star-shaped homo-pentamer that covers the open end of the filament and helps the assembly of FliC [37,38,39,40]. FliD was originally found to be an inhibitor of flagellin polymerization in vitro [41] but it is required for the growth of flagellin in vivo [29]. The analysis of the flagellum structure in *B. melitensis* by transmission electron microscopy (TEM) has identified several characteristics of the sheathed flagella of other species, although little is known about flagellar sheaths in bacteria; indeed, *Brucella* is the only rhizobium that produces a sheathed flagellum [42,43]. Given the sheath surrounding the filament, the visible flagellum of *B. melitensis* has a diameter of 50 nm, which is larger than that of a sheathless flagellum. However, the diameter of the bacterial filament is usually about 20 nm but we show that the diameter of the inner filament in the sheath of *B. melitensis* is only 11 nm. Sheath production is not related to flagellar assembly in *B. melitensis*; this is because in bacterial mutants for structural flagellar proteins (Δ*fliF*, Δ*flgE* and Δ*fliC*), a filamentous appendage is still produced, despite the absence of FlgE or FliC proteins. However, the persistence of an empty sheath in flagellar mutants has often been described in bacteria that produce a sheathed flagellum, such as Vibrio species and *H. pylori* [44,45]. In this light, the flagellum sheath of *B. melitensis* is an extension of the outer membrane containing LPS, which is also observed in *H. pylori*, *B. bacteriovorus,* and some Vibrio species [46,47,48,49]. 

This process of conformation of the flagellar structure has been widely investigated in *E. coli* and *Salmonella*, being a regulated process where genes that are closely linked to the order of the formation of the flagella (class I, II, and III genes) are activated in tandem [50,51]. FlhDC is a master regulator, present in *E. coli* and *Salmonella,* that controls the expression of class II genes [52]; the products of these genes are structural proteins that comprise the basal body and hook as FliF and FlgE, respectively [53]. Likewise, the regulators of these class III proteins in *E. coli* are FliA (a sigma factor, also called σ28) and its anti-sigma, FlgM [32,50]. Once the structure of the hook is completed, the FlgM protein (anti-sigma) is secreted through its channel, which allows σ28 to activate the expression of class-III genes, including FliC, ending the formation of the filament [54].

## 3. Molecular Mechanism of Flagellar Expression in *Brucella*

Although the *Brucella* genome contains flagellar operons, there is no evidence of motility [55]. This information is quite interesting since it was previously reported that under specific growth conditions, *B. melitensis*, being non-motile, was able to form a flagellar appendage by a mechanism that has not yet been clarified [46], and recently, *Brucella* strains isolated from *Ceratophyrus adornada* frogs showed a phenotype with high motility [56]. In the genomic sequencing of isolated non-motile *Brucella*, the presence of non-functional flagellar genes has been found [57], while the analysis of genes encoding flagellar proteins in a recently described motile strain (B13-0095) revealed that all genes were fully functional [56], which could mean that these genes are expressed under environmentally selective pressure [55]. The action of certain bacterial enzymes participates indirectly to fulfill the purpose of achieving the replicative niche. An example of this is the nitrate reductase enzyme of *Brucella*, which is expressed basally in aerobic conditions but increases its expression in hypoxic conditions. This enzyme actively participates with other proteins, such as superoxide dismutase copper-zinc, (SOD Cu/Zn), performing adaptive functions against redox changes that may affect bacterial nidation during infection [58], a very important phenomenon in the survival of *Brucella* when it colonizes an organ, such as the uterus, in infected mammals [59]. This condition, wherein the presence of reactive oxygen species (ROS) increases, activates certain proteins such as MucR (a Ros-like regulator) in *B. melitensis*, *B. suis,* and *B. abortus*, which are also involved in the activation of flagellar genes in response to environmental stress [60,61]. The genes involved in the formation of the bacterial flagellum (*E. coli*; *Salmonella* spp.) differ in certain aspects from those described by Soler-Llorens, such as the location in the genome, or the size of genes and their regulation; however, they maintain the order of the assembly process of proteins of different classes and their regulatory proteins [56] (Figure 1). 

In *Brucella*, it has been suggested that certain proteins participate in this regulatory cascade of flagellar genes, with certain genes coding for the sigma factors RpoD, RpoN, RpoH1, RpoH2, RpoE1, and RpoE2, where the deletion of the *rpoE1* gene increases the production of the flagellar protein FlgE [57]. In addition, this mutation also increases the gene expression of *fliF*, *flgE*, *fliC*, *flaF,* and *flbT* (Table 1), where the expression of these genes is also controlled by the master regulator *ftcR*, which is mediated by the *rpoE1* gene product and other, as yet unknown, proteins [62].

Moreover, the mutation of *rpoE1* increases the promoter activity of the flagellar master regulator *ftcR*, suggesting that the protein group with RpoE1 acts upstream of *ftcR*, repressing its expression. FtcR is a flagellar two-component regulator described in *Brucella melitensis*, which is involved in the expression of class II genes, *flg*E (hook proteins) and *fli*F (basal body proteins) [57,68], as well as certain proteins that regulate the expression of the class-III gene, such as FlaF, which represses FliC, and FlbT (Figure 2A). In addition to being a self-regulating protein, FlbT enables the production of the FliC protein, forming part of the expression of class III genes [70]. FliC production in *Brucella melitensis* is not subject to the basal body and hook termination, unlike *E. coli* and other bacteria [71,72]. In *B. melitensis,* the mutant for FlgE and FliF still produces flagellin, as previously indicated; therefore, the expression of FliC requires the FlbT regulator for its expression [71]. Genes have been shown to be involved in quorum sensing (QS) in *Brucella*; *vjbR* and *blxR* work as transcriptional regulators involved in the virulence of *Brucella* and participate in the formation of biofilms through the expression of flagellar genes [73]. VjbR is required by *B. melitensis* for transcription of the type-IV secretion system and the expression of several flagellar genes (*fliF*, *flhA*, *motB,* or *flgE*) that contribute to its virulence in mice, where *ftcR* is partially activated by VjbR [46,74] (Figure 2B). New studies have identified that FliC production is not controlled solely by the master regulator, FlbT. Along with the abovementioned components, we must mention YbeY endoribonuclease, one of the best-conserved enzymes in different organisms, as it is related to an important variety of metabolic activities including, for example, proper cell morphology, mRNA transcription levels, and virulence, as well as indirectly participating in the expression of certain flagellar proteins in *B. abortus* [75,76]. When the *ybeY* gene was mutated in *Brucella abortus*, it was found that the product of this gene participates indirectly in the gene expression of transcriptional regulatory proteins, since *ftcR* mRNA levels were elevated in the mutant strain. This is linked to the expression of FliC (flagellin), which is also significantly elevated in the Δ*ybeY* strain [77], and, like FtcR, is required for FliC production. The increase observed in the expression of *fliC* mRNA in the Δ*ybeY* strain could be due to the increase in FtcR levels (Figure 2C) [46,68].

From the above information and added to the context of *Brucella* in the literature, it can be stated that the action of flagellar genes is not only subject to the expression of master regulators but is also related to the expression of other genes belonging to other bacterial metabolic cascades. This is clear from different studies conducted on mutant strains of different genes in *Brucella* (Table 2).

## 4. Flagellar Proteins as Virulence Factors in *Brucella* spp.

Virulence factors are molecules produced by pathogens that favor their infective capacity, allowing them to colonize host tissue, evade the host immune response, enter and exit cells, or acquire nutrients (e.g., iron). These are key elements in the adaptation and survival of pathogens in the host organism [84]. The most important characteristic of *Brucella* is the ability to survive and multiply within both phagocytic and non-phagocytic cells [85]. There is also evidence that certain proteins that make up part of the flagellum participate in mechanisms that allow the bacterium to establish its replicative niche and its infective chronicity in the host [57,63,64,68,69]. These flagellar virulence factors have even been studied as immunogenic agents in murine models for the development of brucellosis vaccines [67]. Bacteria of the genus *Brucella* have been described for some time as non-motile organisms [1]; however, certain findings reveal the presence of operons that contain flagellar genes [86], and, in certain species of *Brucella*, under special conditions—either cultivational or environmental—functional bacterial flagella are present [46,87]. For the induction of the disease, as indicated previously, *Brucella* must adhere, invade, and survive within mammalian cells. As previously mentioned, the flagellar genes of *B. abortus,* present in its genome, are an interesting finding in relation to a bacterium described as non-motile [86]. This discovery may indicate that these proteins are involved in other vital functions for the establishment of bacterial infection, as in the case of the FlgJ protein, which, when encoded by the open reading frame BAB1_0260, participating in virulence and being mutated, limits the ability to form biofilms in vitro [69,88]. In *E. coli*, *Salmonella,* and other bacterial groups, this flagellar conformation and its subsequent formation begin with the action of the FlgJ protein. FlgJ is the rod cap protein most closely related to the early stages of flagellum formation and rod assembly; once the latter is formed, FlgJ is cleaved from the emerging flagellum and is essential for its genesis since it has dual canonical activity [89], hydrolysis on the PG and scaffold activity, enabling the assembly of the initial units of the flagellum in the periplasmic space [90]. It has been shown in *S. enterica* that the FlgJ muramidase plays an important role in flagellum synthesis, being secreted through the type-III flagellar export system and exhibiting its enzymatic activity in the C-terminal end [89,91]. This domain, of approximately 164 amino acids in length, presents two important catalytic residues for this muramidase activity. This is given by glutamic acid (E233) and aspartic acid (D248), the function of which is to make a hole in the PG, thus allowing the penetration of the shaft through the periplasmic space so that the hook and the filament can later assemble in the extracellular space [91]. It has been reported, however, that in *S. enterica*, mutants for *flgJ* maintain the ability to swim after long incubation times, which indicates that muramidase activity is not entirely necessary for flagellum formation, suggesting that flagellum formation is due to the assembly upon biogenesis of new membranes and new PG [92]. Likewise, in *Rhodobacter sphaeroides*, SltF is present, a muramidase associated with the flagellar system. Unlike the FlgJ of *S. enterica*, it is transported to the periplasm by the Sec translocase pathway, with the ability to produce a hole in the PG layer for the subsequent penetration of the flagellar structure [93,94]. In the case of *Caulobacter crescentus*, a bacterium that has a cell cycle very similar to *Brucella* [95], it possesses PleA, an enzyme homologous to lytic transglycosylase. This protein contains a region that is similar to a peptidoglycan-hydrolytic active site; a point mutation at this site in PleA results in the loss of the flagellum and pilus biogenesis. Furthermore, PleA is required for flagellar assembly, indicating its involvement during flagellum formation [96]. *B. abortus* has a flagellar protein with hydrolase activity on peptidoglycan, the FlgJ protein (BAB1_0260) located in genomic island three (GI-3) (https://www.ncbi.nlm.nih.gov/protein/82615263, accessed on 15 November 2021), which may be involved in the virulence of this species [69]. Although the muramidase activity has not yet been studied in *Brucella*, its activity in that part of the PG is doubtful; however, the bioinformatic bases suggest an important role for FlgJ in the initial stages of flagellar genesis, due to its glucosaminidase activity (https://www.kegg.jp/dbget-bin/www_bget?bmf:BAB1_0260, accessed on 15 November 2021). A comparative analysis of the amino acid sequence encoding FlgJ from *Brucella* with other bacteria reveals that the FlgJ from *Brucella* does not have the typical conserved sequences of FlgJ with muramidase activity found in *E. coli* and *S. enterica* (Figure 3). Even so, according to some databases, FlgJ from *Brucella* presents activity on the PG (acetylglucosaminidase activity). Like other FlgJ from other bacteria, it could be hypothesized that an extra element might be participating in this muramidase activity, to make a hole in the peptidoglycan before the assembly of proteins in the flagellum. This occurs in *Rhodobacter sphaeroides* or *Caulobacter crescentus*, as mentioned above, where other PG lytic enzymes would participate, together with the FlgJ from *Brucella*. This would, therefore, enable the correct assembly of the base of the flagella in the bacterial membrane. By obtaining mutants for the BAB1_0260 in *B. abortus*, it was possible to evaluate whether FlgJ was involved in pathogenicity as a virulence factor, showing that its deletion is not lethal to bacterial survival [69,97,98]. The FlgJ hydrolyzing activity on peptidoglycan (PG) plays an important role in the growth of this bacterium, specifically during the remodeling of PG during its cell division, which would explain why the elimination of the *flgJ* gene significantly reduces its growth. The bacterial flagellum is associated with functions that differ among bacteria. In addition to motility, it may be related to adherence and biofilm formation [99,100], processes closely related to the chronicity of the infections caused by different bacterial groups [101,102]. Biofilms are extracellular polymeric elements (EPS) produced by microorganisms, which include polysaccharides, proteins, nucleic acids, and lipids, which enable their adhesion to various surfaces and the interaction among bacteria (cell communication, competition, cooperation, or the horizontal movement of genes) [103]. The molecular mechanisms of how *Brucella* generates biofilms and the participating elements have scarcely been explored, since their replicative niche is cells in living tissue, which makes their study in this area difficult. This finding shows that FlgJ participates in the adherence or secretion of proteins involved in the process, making it clear that in the formation of biomass associated with *B abortus* 2308 biofilms, FlgJ is directly related to the levels of bacterial growth and division [69]. This suggests that reduced bacterial adherence decreases the colonization of host tissues, where several factors are involved, including the flagellum [104]. Likewise, microarray assay analyses have found that the effect of erythritol on the expression of *B. melitensis* genes (cultivated with or without erythritol) was shown to be the upregulation of two main virulence pathways, in response to erythritol exposure—the VirB type IV secretion system and flagellar proteins (the third-largest cluster of orthologous groups), suggesting a role for erythritol in virulence [82].

## 5. Evolutionary Aspects of *Brucella* and Flagellar Genes

One of the orders that fall within the class of Alpha-proteobacteria is the order of the Rhizobiales; among the families of this order are the Brucellaceae, with genera such as *Mycoplana*, *Ochrobactrum* and *Brucella,* and the family Rhizobiaceae, which includes the genera *Allorhizobium*, *Azorhizobium*, *Bradyrhizobium*, *Mesorhizobium*, *Rhizobium* and *Sinorhizobium.* The genus *Brucella* developed as intracellular animal pathogens, while the genus *Rhizobium* is associated with soil and forms nitrogen-fixing nodules but, according to this evolutionary background, they have similar evolutionary characteristics [105]. Some evidence to hypothesize an evolutionary relationship between Brucellaceae and Rhizobiaceae is given by the presence of certain genes that they have in common; for example, some genes involved in symbiosis in *Sinorhizobium meliloti* are homologous to genes involved in *Brucella* pathogenesis [106]. The *bacA* gene that is involved in the symbiosis of *Sinorhizobium meliloti* has its homologous gene in *Brucella abortus*, which plays a role in the survival of *Brucella* in macrophages and in its pathogenesis in mice [106,107]. Along with the above findings and in relation to the bacterial flagellum, we should mention that *Bradyrhizobium diazoefficiens*, the nitrogen-fixing symbiont of soybean, possesses two flagellar systems that evolved independently [105,108,109]. This dual flagellar system, present in such bacteria, performs its function as a single flagellum, which is similar to the polar flagellum of *Brucella* spp. [109,110]. A comparative analysis of the genetic sequences encoding the FliC protein of *Brucella abortus* 2308 flagella revealed that bacteria of different genera in the order Rhizobiales show high homology of sequences and structural domains of the FliC protein (Figure 4). This is in conjunction with the consideration that a high percentage of bacteria of the Rhizobiales order are considered to be environmental microorganisms and present bacterial flagella [111,112,113].

Thus, we could speculate that *Brucella* diverged from common ancestors that are generally associated with soil, which are plant symbionts; that, like *Brucella*, they are intracellular and, therefore, do not require the expression of flagella. However, if the genes that encode the vast majority of the constituent proteins of a flagellum are subjected to selective pressure from their environment, the flagellum can be expressed [46,108]. Furthermore, all this is closely related to the proposition that *Brucella*’s ancestor was probably a free-living plan-related bacterium, possessing one chromosome and evolving into an animal parasite with two chromosomes [114] (Figure 5).

Within this evolutionary process, *Brucella* has acquired different foreign DNA fragments by horizontal transfer, which are distributed throughout the genome, and which encode several proteins, mainly of unknown function [115]. Yet, among these genes, there are some that encode virulence factors, where some flagellar proteins that are present in certain *Brucella* ORFs are worthy of note [69,86,116,117].

## 6. Flagellum and Immune Response against *Brucella*

The function of flagella as bacterial nanomachines is to mediate motility. The flagellar filament, composed of a single protein, flagellin, is assembled by the basal body via a pathway homologous to the type 3 secretion system (T3SS) [118]. In motile bacteria, such as *Bartonella* species (lineages 1–3, except *Bartonella bovis*), flagella are thought to mediate erythrocyte internalization as a mechanical force or as adhesion molecules [119]. The *Bartonella* species of lineage 4 that are non-motile, due to the loss of flagella, mediate interaction with erythrocytes through Trw T4SS, suggesting that Trw has functionally replaced the flagella in establishing interaction with erythrocytes [120]. *Brucella* are non-motile bacteria but flagella genes are essential for their virulence [121]. Given that *Brucella* is also found in erythrocytes [122], we could speculate that there is a mechanism similar to the one mentioned earlier, where flagellar genes play a fundamental role in the survival and stealth of *Brucella*, even, one could say, in the evasion of the immune response. *Brucella* spp. evade detection through TLR4 by producing a poorly recognized form of lipid A. *Brucella* lipid A contains a much longer fatty-acid residue (C_28_) than enterobacterial LPS (C_12_–C_16_), and this modification greatly decreases its endotoxic properties by reducing TLR4 agonist activity and, thus, the host immune response [123]. In addition to TLR4 evasion, *Brucella* evades detection by TLR5 by producing a flagellin that lacks the TLR5 agonist domain [124]. As a result, *Brucella* flagellin is a poor inducer of TLR5-mediated inflammatory responses [123]. In addition to manipulating T cell-mediated immune responses, *Brucella* can also control T cell-independent immune responses [125]. Therefore, it is likely that *Brucella* induces partial and transient immunosuppression to establish a chronic infection [126,127]. Conversely, certain flagellar proteins have been examined for their possible immunogenic value as vaccine candidates. Five flagellar genes (BAB1_0260 (FlgJ); BAB2_0122 (FliN); BAB2_0150; BAB2_1086; BAB2_1093) were taken into consideration for their ability to induce humoral and cellular responses and to protect mice against *B. abortus*. FlgJ and FliN proved to be protective antigens that produced humoral and cellular responses in mice [128]. In addition to the above findings, studies with the FliC protein have shown it to be a new potential antigen candidate for the development of a subunit vaccine against *Brucella* [129,130].

## 7. Conclusions and Outlook

In recent years, many new species of *Brucella* have been discovered, mainly pathogens of marine mammals and others capable of infecting terrestrial mammals, in addition to flagellated species present in amphibians [86]. This complicates the execution of control programs since the recently characterized *Brucella* species have high genetic flexibility and many of these isolates are motile, fast-growing, able to survive in the soil, more resistant to unfavorable conditions of environmental acidity, and more able to adapt to new non-mammalian hosts, such as amphibians, rapidly adapting to their environment to broaden their host range [87]. As *Brucella* is a pathogen with different routes of infection, it can survive in and out of mammalian hosts for a long time, even under unfavorable conditions. *Brucella* is characterized as being a stealthy microbe that tends to chronicity, instead of causing an acute fatal infection, standing out for being a successful microorganism by evading the immune response and keeping its hosts alive to maintain its own survival. Flagellar biogenesis shows interesting variations among different proteobacteria and even among members of Alphaproteobacteria. The flagellar genes of *Brucella* are needed to establish infection in vivo in mice; however, the molecular basis of the impact of the flagellum on virulence in *Brucella* is still to be determined. This is evidenced by certain species of *Brucella,* such as *B. melitensis* 16M, *B. ovis*, *B. abortus* 2308, and certain *Brucella* spp. in amphibians expressing flagellar proteins and/or flagella at the intracellular level [46,64,68,69,87]. This is an event of great importance, considering that today there is a strong relationship between humans and farm animals, which will eventually lead to an increase in this zoonosis caused by *Brucella*. If we pay attention to the routes of infection that *Brucella* has, in relation to its scope in the mechanisms of pathogenesis and transmission of the infection, we may hypothesize its relationship to the possible expression of the flagellum during invasion, and its persistence in red blood cells in the murine model [122]. This phenomenon is closely related to the role that the flagellum could have during its residence inside blood-sucking insects [131]. With all the above, we could hypothesize that the flagellum is playing an important role in bacterial survival at the extracellular, intracellular, or environmental level. The study of the bacterial flagellum in a non-motile bacterium is undoubtedly a challenge that deserves attention since it seems to be directly involved in its survival in different hosts and in a hostile environment.

Some methodologies have been proposed in the literature to explain the importance of flagellar genetic sequences involved in the abovementioned events. The creation of mutants, the study of biofilm formation, and the infective capacity both in vitro and in vivo [69] are part of the most successful methodology for preliminary assays. Likewise, the study of flagellar protein translocation by different types of transporters, where the role of these proteins in *Brucella* virulence could be evaluated [132], might point out certain functional evidence between different bacterial structures. Flow cytometry analysis in infected cells, to discern the appearance of markers involved in the immune response [130], as well as immunoprecipitation assays to see eventual protein interactions [130], and simple experiments such as staining of the flagella at different growth times under a variety of conditions [66] would provide further data to clarify certain unknowns about flagellar function and structure. Furthermore, in the field of immunology, the preparation of vaccines of different flagellar proteins in search of potential immunogenic agents [133], and the use of bioinformatics tools for comparative study with other bacteria [134,135], with differing degrees of similarity in the mechanisms of infection are appropriate methodologies to gain more background information. This will allow us to understand more fully the structure–function relationship of the flagellum in an organism described to date as a non-motile bacterium.

## Figures and Tables

**Figure 1 microorganisms-10-00083-f001:**
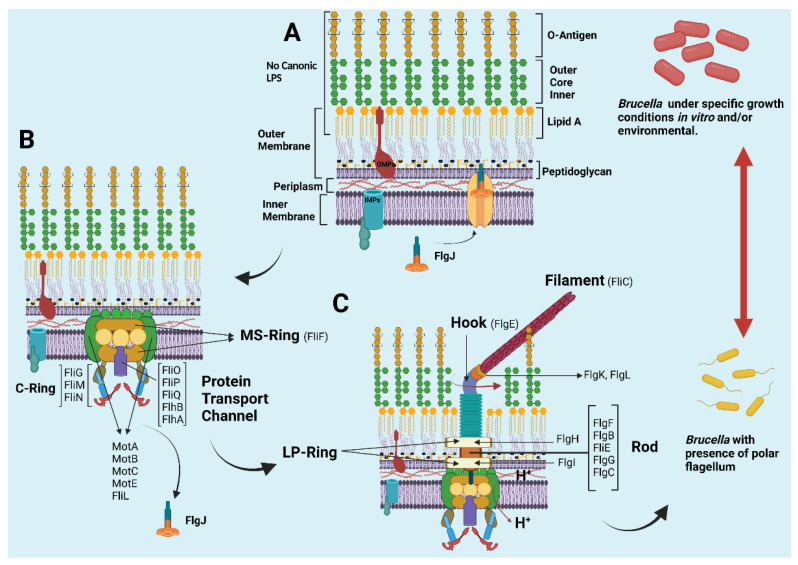
Diagram of the bacterial flagellum of *Brucella*, according to the classical structure of the flagellum in Gram-negative bacteria and the most up-to-date literature. *Brucella* expresses its bacterial flagellum under specific and controlled in vitro growth conditions, and in the face of different environmental changes, the expression of the flagellum is involved in its survival in a hostile environment when it does not have a definitive host. (**A**) Under these specific conditions and after expression of the master regulators at gene level, FlgJ acts by drilling the membrane and the peptidoglycan (PG), a key protein in the construction of the flagellum base structure. (**B**) The correct functioning and conformation of this bacterial organelle depend on the perfect arrangement of the proteins that form the MS-ring, the transporter channel proteins, and the C-ring. Once the base of the flagellum has formed on the membrane, FlgJ is cleaved from the basal body. (**C**) This correct arrangement of previous proteins makes it possible to reach the end of the flagellum formation given by the cluster conglomerate of rod proteins, the LP-ring, followed by the hook, and finally, by the formation of the filament (FliC) that mediates the movement allowed by proton pumping from the periplasm. In Gram-negative bacteria, the final protein that “seals” the filament is called CapD (FliD); however, this has not yet been described in *Brucella*, even though *Brucella* presents the gene for its expression. LPS: lipopolysaccharide; OMP: outer membrane protein; IMP: inner membrane protein.

**Figure 2 microorganisms-10-00083-f002:**
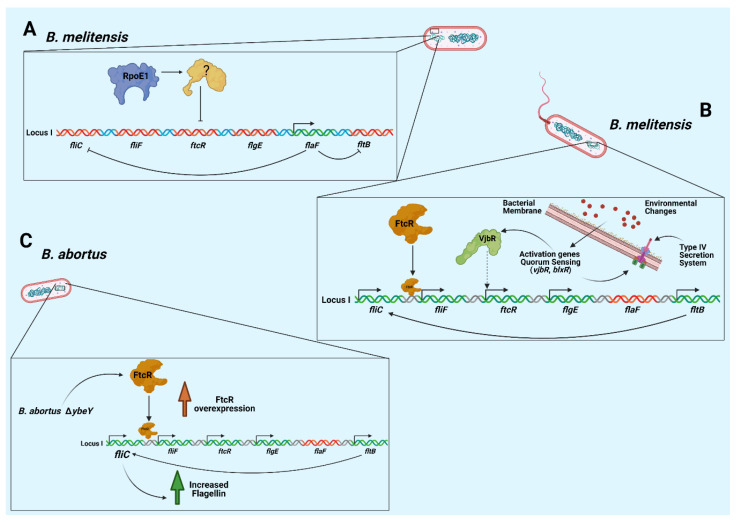
Transcriptional regulation of some of the flagellar genes described in *Brucella* at locus I. (**A**) Repression of flagellar expression in *B. melitensis* is linked by the repressor RpoE1 (and other unknown proteins) to *ftcR*. This master regulator allows FlaF to be expressed and inhibits the expression of *fliC* and *fltB.* (**B**) The expression of a unipolar flagellum in *B. melitensis* is linked to FtcR, a direct regulator of flagellar protein expression; this is partially activated by the regulator VjbR, which is involved in quorum sensing and the expression of the type-IV secretion system. (**C**) In *Brucella abortus,* when the *ybeY* gene (BAB2_1156), an endoribonuclease, is deleted, there is an overexpression of the FtcR and this results in an increase in flagellin.

**Figure 3 microorganisms-10-00083-f003:**
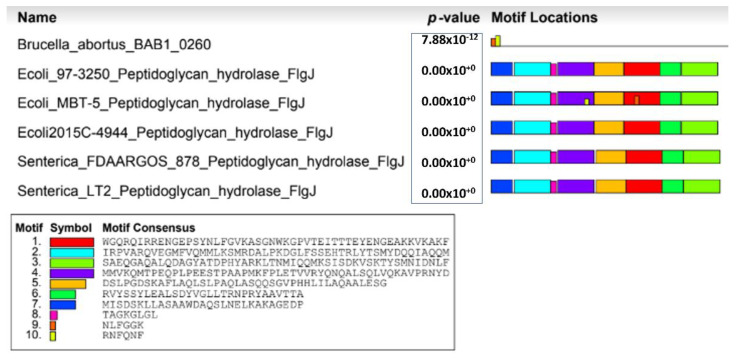
*p*-value of conserved motifs related to *flgJ* gene of *Brucella abortus* 2308. Bioinformatics tool used: http://meme-suite.org (accessed on 15 November 2021).

**Figure 4 microorganisms-10-00083-f004:**
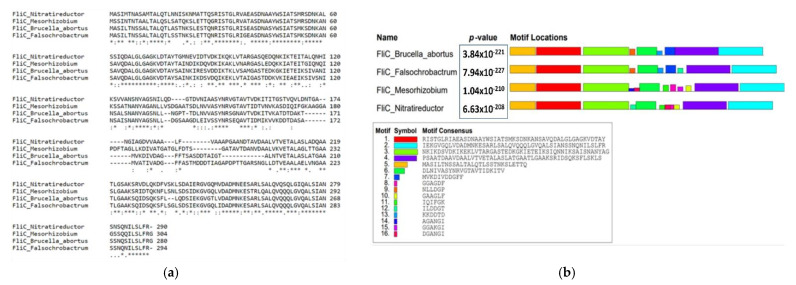
Bacterial FliC amino-acid sequence analysis. (**a**) Multiple alignments of sequences of the FliC protein of *Brucella abortus* (Q2YJF1) against the FliC protein of *Falsochrobactrum* sp. HN4 (A0A316J970), *Mesorhizobium* sp. UASWS1009 (A0A1C2EA77) and *Nitratireductor pacificus* pht-3B (K2M4Y7) (**b**). There are 16 conserved motifs in relation to the *fliC* gene present in the abovementioned species. Bioinformatics tool used: http://meme-suite.org (accessed on 15 November 2021).

**Figure 5 microorganisms-10-00083-f005:**
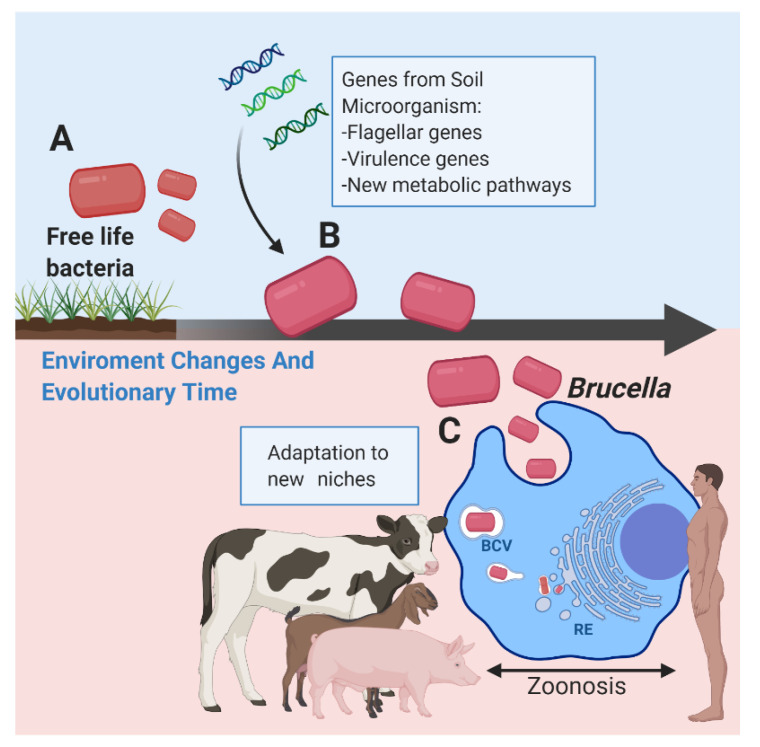
(A) In terms of its evolutionary origin, it is postulated that *Brucella* may have been a free-living bacterium. (B) Over time, and according to its evolution, it remained in contact with other microorganisms, such as soil bacteria and fungi, managing to acquire certain genes that, therefore, improved its metabolic resources. (C) Environmental changes helped give it the ability to adapt to new hosts, such as eukaryotic cells. In *Brucella*, certain virulence factors that are different from other bacteria, like its LPS, its type-four secretion system (T4SS) and the BvrR/BvrS system, enable *Brucella* to interact with the host cell surface and form an early *Brucella*-containing vacuole (BCV) for its subsequent interaction with the endoplasmic reticulum (ER), where bacteria multiply and reach their replicative niche. Likewise, the presence of flagellar genes and the active participation of flagellum proteins in different functions, such as the translocation of proteins to the outside and the formation of adhered biofilm-associated biomass gave *Brucella* the ability to be a successful organism in achieving its infectious chronicity and evading the immune response of the host organism.

**Table 1 microorganisms-10-00083-t001:** The gene codes of *Brucella abortus* flagellar proteins are presented; in some cases, they are described as pseudogenes and have not been studied.

Flagellar Structure	Proteins	Function	Gene Code	Location	Reference
Filament	CapD (FliD)	Term protein in the structure of the flagellum, adhesion to mucin.	BAB1_0534	Distal end of filament	Database Kegg Genes
	FliC	Filament subunits, immunity, coadjuvant in others species.	BAB2_1106	Filament	[63]
	FlgL	Proteins that adhere the flagellum to the hook.	BAB2_1096	Initial base of filament	[64]
	FlgK		BAB2_1097	External OM	[64]
Hook	FlgE	Hook protein.	BAB2_1098	External OM	[64]
Rod	FlgH	Structural proteins that form the L-ring.	BAB2_0156	External OM	[64]
	FlgI	Structural proteins that form the P-ring.	BAB2_0153, BAB2_0154	PG	[64]
	FlgG	Structural proteins that form the channel.	BAB2_0151	OM/PG	[64]
	FlgF		BAB2_0127		[64]
	FlgC		BAB2_0149		[64]
	FlgB		BAB2_0148		[64]
	FliE		BAB2_0122		[65]
External position Export Gate	MotE	Rotor stabilizer proteins.	unknown	Intermembrane space	N.Y.I.B.*
	MotC		BAB2_1102	Intermembrane space	[64]
	MotA		unknown	Intermembrane space	N.Y.I.B.*
	MotB		BAB2_1103	(PG, PS, IM)	[46]
MS-ring	FliL	Protein that binds Mot structures to the MS-ring.	pseudogene	(PG, PS, IM)	N.Y.I.B.*
	FliF	Structural proteins that form the MS-ring.	BAB2_1105	IM (MS-ring)	[66]
C-ring	FliG	Structural proteins that form the C-ring.	pseudogene	Cytoplasm	N.Y.I.B.*
	FliM		pseudogene	Cytoplasm	N.Y.I.B.*
	FliN		BAB2_0122	Cytoplasm	[67]
	FliH/FliI/FliJ	Primary role: flagellar proteins of the export apparatus. Secondary role: Stabilizing proteins of the C-ring.	pseudogene	Cytoplasm	N.Y.I.B.*
Export Gate	FliO	Structural proteins that form the export gate. Type III secretion exporter	pseudogene	IM	N.Y.I.B.*
	FliP		BAB2_0158		[64]
	FliQ		BAB2_1092		[64]
	FliR		BAB2_1088		[64]
	FlhA		BAB2_1089, BAB2_1091		[46]
	FlhB		BAB2:0120		[46]
Regulator Genes	FtcR	Transcriptional regulators.	BAB2_1099	Cytoplasm	[68]
	FlaF	Transcriptional regulators.	BAB2_1095	Cytoplasm	[62]
	FlbT	Transcriptional regulators.	BAB2_1094	Cytoplasm	[55]
	FliK	Molecular ruler for hook length control.	pseudogene	Cytoplasm	N.Y.I.B.*
	FlgD	Cap foldases for hook.	BAB2_1093	Cytoplasm	[64]
	FlgN	Chaperone for FlgK.	pseudogene	Cytoplasm	N.Y.I.B.*
	FlgA	Chaperone for FlgI.	BAB2_0152	Cytoplasm	[64]
	*fliR gene*	Biosynthesis of FliR.	BAB2_1088	Bacterial export protein	[64]
	FlgJ	Mannosyl-glycoprotein endo-beta-N-acetylglucosamidase in PG	BAB1_0260	Cytoplasm/Inner membrane	[69]

N.Y.I.B.*: not yet identified in *Brucella*.; OM: Outer membrane; PG: Peptidoglycan; IM: Inner membrane; PS: Periplasmic space.

**Table 2 microorganisms-10-00083-t002:** Summary of the effects related to the mutation of genes encoding flagellar proteins or genes, related to their expression in *Brucella*.

Mutant	*Brucella* Species	Effect	References
Δ*ftcR*	*B. melitensis*	Decreases flagellar gene expression.	[68]
Δ*vjBR*	*B. melitensis*	Decrease flagellar proteins expression.	[74,78,79]
Δ*blxR*	*B. abortus*	Regulates virulence factors (TSSIV and flagella).	[73]
Δ*bvrR*/*bvrS*	*B. abortus*	Decreases virulence.	[80]
Δ*ftcR*, Δ*fliF*, Δ*flgE*, Δ*fliC*	*B.melitensis*	Empty sheath.	[62]
Δ*flgJ*	*B. abortus*	Decreases biofilms, virulence and infection capacity.	[69]
Δ*ybeY *(BAB1_2156)	*B. abortus*	Multiple dysregulations of expression of bacterial proteins, including proteins and flagellar regulators.	[77]
Δ*fliF*, Δ*flhA*, Δ*motB,* Δ*flgE*	*B. melitensis*	Dysfunction is the first step of infection or cycle of life.	[46]
Δ*aibP (QS gen)*	*B. melitensis*	Expression of *fliC*, *fliF*, *flgE* y *flbT* was significantly downregulated, with enhanced *virB* genes expression and VirB8 production.	[63]
ΔBM-LOV-HK (light-sensing histidine kinase)	*B. melitensis*	Sigma factor *rpoE1* downregulated, with flagellar, quorum sensing (QS). Type-IV secretion system genes were upregulated.	[81]
Δ*bpdA*	*B. melitensis*	Downregulation of flagellar promoter activities. Attenuation of virulence. Increase in biofilms.	[82,83]
Δ*RpoE1*	*B. melitensis*	Overexpresses the flagellar protein FlgE. Increases promoter activity of the flagellar master regulator *ftcR.*	[62]

## Data Availability

https://www.ncbi.nlm.nih.gov/protein/82615263, visualized based on data from the website https://pubmed.ncbi.nlm.nih.gov/. https://www.kegg.jp/dbget-bin/www_bget?bmf:BAB1_0260, visualized based on data from the website https://www.genome.jp/ (all accessed on 15 November 2021).

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
