# Peer review of "Brucella and Its Hidden Flagellar System"

_microorganisms, 2021, doi:10.3390/microorganisms10010083_

Round 1

Reviewer 1 Report

This is a very interesting topic and a comprehensive review of the research literature related to the topic. Below are a list of specific comments for the authors' consideration.

Line 48 – not sure what the authors mean by “through rapid acidification of the medium”. What is the medium? The inside of the phagosome? Cytoplasm of the phagocyte?

Line 50 – delete phrase “successfully achieves its objective”. This is anthropomorphizing the bacterium.

 Lines 56-60 – This is a run-on sentence that is difficult to follow. Consider breaking up into two sentences.

Lines 76-80 – The description of the flagellar motor is difficult to follow and somewhat inconsistent with what is stated later. I am not sure what is meant by the statement “allowing the interaction of structural proteins (FliE, FlgB, FlgC, FlgF). In the sentence starting on line 82, the authors indicate FlgB, FlgC, and FlgF are part of the ‘tubular axial structure’. There is no mention of the flagellar protein export apparatus associated with the MS-ring. Authors should indicate the L-ring and P-ring function as a bushing for the flagellar motor.

Line 101 – change “the products of these genes encode structural proteins ….” to “the products of these genes are structural proteins the comprise the basal body and hook”

Line 103 – FliA and FlgM are not regulators of Class II gene, but the Class III genes as stated in the next sentence.

Line 117 – The authors were discussing the presence of flagellar genes in Brucella species, and then switch abruptly to the subject of nitrate reductase without indicating how these two topics are related.

Lines 129-135 – Another run-on sentence that is difficult to follow. Consider breaking it up into multiple sentences.

Table 1 – I do not feel the column entitled “ubication” adds much to the information presented in Table 1. A figure showing the location of the proteins would be more helpful.  If the authors choose to keep this information though, I have the following suggestions.  First, I was not familiar with the term “ubication”. “Location” is a term that readers are more likely to be familiar with. For FliD and FliC, it would be better to indicate ‘filament’ instead of ‘flagellum’ since flagellum refers to the entire structure, including basal body, hook and filament. FlgK and FlgE are not associated with the outer membrane, but are external to it. “O.M.: out membrane” should be “O.M: outer membrane”.

Line 144 – Sentence starting “FtcR (a flagellar two—component regulator” is an incomplete sentence. In addition, FlgE and FliF should be indicated as gene designations since it is the “expression of class II genes” that is referred to in the sentence.

Lines 146-149 – The sentence is confusing. There are three clauses in the sentence that begin with “which” that make following the sentence difficult.

Lines 151-154 – Interesting that the B. melitensis flgE and fliF mutants express FliC. Is this FliC exported? I would guess not since the flagellar export apparatus is not likely to be functional in the fliF mutant. Any known role for FliC expressed in the absence of FlgE and FliF?

Lines 155-158. Given that VjbR regulates expression of the Type IV secretion system and several flagellar genes, is there any evidence to suggest the T4SS secretes some of the flagellar proteins?

Table 2 – There are a couple of typos in the table – Eexpression in second line, BlxR and BvrR should be blxR and bvrR. What is meant by “flagellar protein dysregulation”?

The authors indicate in Table 2 that some of the mutants result in empty sheaths. The authors should discuss the flagellar sheath when they are describing the structure of the flagellum. In Table 1 the authors indicate FliD has a role in adherence to mucin. If the filament is encased in a sheath, one would expect this to prevent FliD from being surface exposed and binding mucin. Some discussion of this might be helpful.

Line 253-258. Run-on sentence. Consider re-writing.

Line 317 – Unclear what is meant by “These genes bac involved in differentiation”. Are these the same genes that are referred to in the sentence that precedes this statement?

Lines 338-343. Run-on sentence. Consider re-writing.

Reviewer 2 Report

In this article, the authors have gathered and discussed the information available about the flagellum of the brucella bacteria. They propose several interesting evolutionary hypotheses explaining its presence in a bacterium considered to be non-mobile. The article is well written and interesting to read.

The discussion on the environmental pressures that can lead to the expression of the genes encoding the flagellum could be further enriched. For example, several articles have shown that Brucella can persist in blood-sucking insects. Brucella has also been shown in mice to infect erythrocytes. This suggests that in nature Brucella could have an infectious cycle in animals involving a stage of infection and persistence in the insect. It is possible that in the latter the persistence is not intracellular, for example, and is favored by the expression of the flagellum.

On the other hand, the authors discuss little about the constraints imposed by the host's immune response on the bacteria. In mice, intranasal infection does not appear to be affected by the presence of brucella-specific antibodies, suggesting that the bacteria is able to spread throughout the body from the lung to the spleen, remaining inside the cells. cells. Conversely, during a systemic infection, the presence of specific antibodies allows the elimination of the bacteria. It therefore appears that brucella, during mucosal infection, has a strategy that enables it to gain phagocytic cells very quickly. This brief phase could involve the flagellum. Inactivation of the FliC gene increases the inflammatory response against Brucella in mice. Which suggests that the flagellum may have a role in Brucella's stealth.

In science, it is better to come up with testable and refutable hypotheses. The authors should consider proposing experiments to elucidate the role of the flagellum in brucella and to test their evolutionary hypotheses. This would stimulate research on this subject. 

Minor modification:

1 / In the abstract, Brucella is described as a microorganism, which is rather imprecise. It would be better to replace this term with gram negative bacteria.

2 / As the article describes in detail the structure of the flagellum, a figure of it would be useful 

Round 2

Reviewer 1 Report

The review examines the role of putative flagellar genes in the physiology and virulence of Brucella species, most of which are reported to be non-motile. The topic is of broad interest and the review covers the topic sufficiently. The revisions made by the authors address many of the previous problems with content and writing, but there are still some minor points the authors need to address.

Specific comments:

1. There are several spelling and grammatical errors throughout the manuscript. For example, 'melitensis' is frequently misspelled in the manuscript. Some specific grammatical problems and suggested changes are listed below.

Line 11 – change “including” to “as well as”

Line 13 – change the sentence starting “The bacterial flagellum” to “While motility is the primary role of the flagellum in most bacteria, in Brucella the flagellum is involved in virulence, infectivity, cell growth and biofilm formation.”

Line 27 – change ‘species of bacteria” to ‘bacterium”

Line 79 – I do not know what the authors mean by the statement “together with added to interaction of structural proteins (FliE, FlgB, FlgC, FlgF)”.

Line 132 – remove “DNA”

Line 189 – remove parentheses and change sentence to “FtcR is a two-component regulator described in B. melitensis that is involved in …”

Line 465 – What is meant by “the blood cell level. murine reds”?

The last sentence is an extremely long run-on sentence that spans lines 471-483. Consider revising the sentence to make more readable.

2. In Table 1, the role of FliH/FliI/FliJ is indicated as stabilizing proteins of the C-ring. This may be a secondary role, but these proteins are part of the flagellar protein export apparatus and their primary role is export of axial components of the flagellum. "Rod" is also misspelled in Table 1.

3. Line 310 – The authors indicate the FlgJ is not a structural part of the flagellum in E. coli and Salmonella, which is not completely true. FlgJ is the rod cap protein and associates with the nascent flagellum during rod assembly. Once the rod is formed, FlgJ dissociates from the nascent flagellum.  It is also premature to indicate BAB1_0260 is equivalent to FlgJ based on the homology with known FlgJ proteins. For example, is there biochemical data that shows BAB1_0260 has muramidase? BAB1_0260 significantly larger than E. coli FlgJ (706 versus 313 amino acid residues), suggesting BAB1_0260 has domains/functions not found in FlgJ.  BAB1_0260 does not appear to share homology with the N-terminal domain of FlgJ, which  is believed to bind to the growing tip of the rod as a capping protein and help assembly of the rod component proteins. The lack of homology with the N-terminal domain of E. coli FlgJ does not mean BAB1_0260 is not equivalent to FlgJ (e.g., B. melitensis and E. coli may be too distantly related for their rod assembly domains to have significant homology). Interestingly though, a BLAST analysis of BAB1_0260 failed to identify homologs in flagellated bacterial species closely related to B. melitensis (Bradyrhizobium japonicum, Rhizobium leguminosarum, and Sinorhizobium meliloti). Taken together, these observations raise serious doubts as to whether BAB1_0260 is equivalent to FlgJ.

Author Response

The review examines the role of putative flagellar genes in the physiology and virulence of Brucella species, most of which are reported to be non-motile. The topic is of broad interest and the review covers the topic sufficiently. The revisions made by the authors address many of the previous problems with content and writing, but there are still some minor points the authors need to address.

Specific comments:

  1. There are several spelling and grammatical errors throughout the manuscript. For example, 'melitensis' is frequently misspelled in the manuscript. Some specific grammatical problems and suggested changes are listed below.
  • was changed as suggested by the reviewer.

Line 11 – change “including” to “as well as”

  • was changed as suggested by the reviewer.

Line 13 – change the sentence starting “The bacterial flagellum” to “While motility is the primary role of the flagellum in most bacteria, in Brucella the flagellum is involved in virulence, infectivity, cell growth and biofilm formation.”

  • was changed as suggested by the reviewer.

Line 27 – change ‘species of bacteria” to ‘bacterium”

  • was changed as suggested by the reviewer.

Line 79 – I do not know what the authors mean by the statement “together with added to interaction of structural proteins (FliE, FlgB, FlgC, FlgF)”.

  • was changed as suggested by the reviewer.

Line 132 – remove “DNA”

- All right, the word "DNA" was removed.

Line 189 – remove parentheses and change sentence to “FtcR is a two-component regulator described in B. melitensis that is involved in …”

  • was changed as suggested by the reviewer.

Line 465 – What is meant by “the blood cell level. murine reds”?

  • was changed as suggested by the reviewer.

The last sentence is an extremely long run-on sentence that spans lines 471-483. Consider revising the sentence to make more readable.

  • The sentence was changed as suggested by the reviewer.
  1. In Table 1, the role of FliH/FliI/FliJ is indicated as stabilizing proteins of the C-ring. This may be a secondary role, but these proteins are part of the flagellar protein export apparatus and their primary role is export of axial components of the flagellum. "Rod" is also misspelled in Table 1.

- was changed as suggested by the reviewer.

  1. Line 310 – The authors indicate the FlgJ is not a structural part of the flagellum in E. coli and Salmonella, which is not completely true. FlgJ is the rod cap protein and associates with the nascent flagellum during rod assembly. Once the rod is formed, FlgJ dissociates from the nascent flagellum.  It is also premature to indicate BAB1_0260 is equivalent to FlgJ based on the homology with known FlgJ proteins. For example, is there biochemical data that shows BAB1_0260 has muramidase? BAB1_0260 significantly larger than E. coli FlgJ (706 versus 313 amino acid residues), suggesting BAB1_0260 has domains/functions not found in FlgJ.  BAB1_0260 does not appear to share homology with the N-terminal domain of FlgJ, which  is believed to bind to the growing tip of the rod as a capping protein and help assembly of the rod component proteins. The lack of homology with the N-terminal domain of E. coli FlgJ does not mean BAB1_0260 is not equivalent to FlgJ (e.g., B. melitensis and E. coli may be too distantly related for their rod assembly domains to have significant homology). Interestingly though, a BLAST analysis of BAB1_0260 failed to identify homologs in flagellated bacterial species closely related to B. melitensis (Bradyrhizobium japonicum, Rhizobium leguminosarum, and Sinorhizobium meliloti). Taken together, these observations raise serious doubts as to whether BAB1_0260 is equivalent to FlgJ.

We greatly appreciate your comments, this has not helped to improve the review and I hope we have been able to answer your questions, for this we have modified and rewritten the article between lines 310 and 347.

Based on your comments, we made some modifications to the text and added more information regarding FlgJ. It was an error on our part to state that E. coli FlgJ was not part of the flagellum. On the other hand, as the reviewer says, there is no experimental evidence that can ensure that BAB1_0260 has muramidase activity. As you mention in your comment, if one compares FlgJ from Brucella to other FlgJ proteins from other bacteria such as E. coli and S. caulobacter, the Brucella protein largely lacks the important conserved functional domains of muramidase activity, for this we include a new figure (Figure 3). On the other hand, it can be assumed that there may be an additional element involved in this muramidase activity before protein assembly in flagella, as occurs in Rhodobacter sphaeroides, in which StlF, a lytic flagellar enzyme that participates together with FlgJ, allowing correct assembly of the  base of flagella in the bacterial membrane With regard to your doubts, we believe that BAB1_0260 may still be called FlgJ, but with only its acteylglucosamidase activity, if one makes a sequence comparison of the BAB1_0260 protein. Regarding your doubts, we believe that BAB1_0260 may still be called FlgJ but only because of its acteylglusoaminidase activity. If a sequence comparison of the BAB1_0260 protein is made with the databases, values ​​close to 60% homology are observed with FlgJ of certain bacteria with peptidoglycan hydrolase activity and with about 40% of those identified with other proteins FlgJ that have muramidase activity. Therefore, as long as the experimental analysis is not available, it is only a name based on sequence identity, which is widely used.